# COVID-19 Vaccine Booster Dose Acceptance among Older Adults

**DOI:** 10.3390/vaccines11030542

**Published:** 2023-02-24

**Authors:** Mehmet Akif Sezerol, Selin Davun

**Affiliations:** 1Epidemiology Program, Institute of Health Sciences, Medipol University, Istanbul 34810, Turkey; 2Health Management Program, Graduate Education Institute, Maltepe University, Istanbul 34857, Turkey; 3Sultanbeyli District Health Directorate, Istanbul 34935, Turkey

**Keywords:** COVID-19, vaccine, booster dose, older adults, acceptance

## Abstract

This study aimed to determine the factors that cause individuals between the ages of 65 and 75 to not receive the third dose of a COVID-19 vaccination, to advise those who are hesitant, and to learn their thoughts about taking the third dose. (1) Material and Methods: This cross-sectional study was conducted between April and May 2022 among 2383 older adults aged between 65–75 who had never received a COVID-19 booster vaccination, according to the records of the District Health Directorate in the Sultanbeyli district of Istanbul. A three-part questionnaire prepared by researchers was given to the older adults via telephone. For statistical analysis of the data, the Chi-square test was used to compare variables; *p* < 0.05 was considered statistically significant. (2) Results: This research was completed with 1075 participants, reaching 45% of people aged 65–75 who did not receive the third dose of the COVID-19 vaccine in the region. In total, 64.2% of the participants were female and 35.8% were male, and the mean age was 69.33 ± 2.88. Those who had previously received an influenza vaccine were 1.9 times (95% CI 1.22–2.99) more likely to seek vaccination. Educational status also played a role, as older adults who were uneducated were 0.5 times (95% CI 0.42–0.76) less likely to seek vaccination. In addition, those who stated that lack of time was the reason for not vaccinating were 1.4 times (95% CI 1.01–1.98), and those who did not have it due to forgetting, 5.6 times (95% CI 2.58–12.24), more likely to seek vaccination. (3) Conclusion: This study shows, in detail, the importance of informing older adults, who have not received the third dose of vaccine for COVID-19 and who are in the risk group, as well as those who are not fully vaccinated, about the risks of not being vaccinated. We believe that it is important to vaccinate older patients; further, since immunity conferred by vaccination may decline over time, mortality rates decrease with the administration of additional doses.

## 1. Introduction

The proportion of the population over the age of 65 is gradually increasing throughout the world, and is becoming more susceptible to infectious diseases, along with the prevalence of chronic diseases. The susceptibility to infectious diseases in people aged 65 and over is increasing due to reasons such as chronic comorbidity, increased use of immunosuppressive drugs, and weakened cellular and humoral immunity [1]. The fact that COVID-19 presented a higher mortality rate among older individuals during the pandemic period has revealed that older adults constitute a population that is sensitive to COVID-19 infection, as with other infectious diseases. As a public health approach, vaccination against vaccine-preventable diseases is of great importance, in accordance with the principle of “protection is superior to cure” [2].

The high contagiousness of the COVID-19 virus and the limited treatments that can improve the prognosis of the disease increase the role and importance of vaccination [3]. Although the first COVID-19 vaccination studies began at the end of 2020, it is the Coronavac vaccine of the Sinovac company produced in China that first began to be implemented in Turkey, and the vaccination program was initiated for the priority group of healthcare workers and older adults. The second vaccine to be implemented in Turkey was the Pfizer BioNTech COVID-19 vaccine, an mRNA vaccine [4]. This was the first COVID-19 vaccine to be approved for both emergency and regular use by the WHO. The continued emergence of SARS-CoV-2 variants with decreased immunity and immune-evasion potential following infection or vaccination has indicated that booster doses of COVID-19 vaccines are required [5]. In the context of COVID-19 vaccination, the 3rd dose of the vaccination is recommended between 4 and 6 months following the 2nd dose based on available evidence showing that protective immunity is reduced 4–6 months after primary vaccination [6,7]. The COVID-19 vaccine is not mandatory as other vaccines are. There is no obligation for older adults, who are at the highest risk of COVID-19 transmission in the population groups of society, to get vaccinated. According to the latest data from the Ministry of Health, while the rate of adults over the age of 18 who received the second dose of the vaccine was 85.6%, the rate of those who received the third dose of vaccine was only 30% [8]. It has been shown that the rate of uptake of a COVID-19 booster dose among the general population varied in different countries, ranging from 62–67% in the USA and 67–71% in Poland to 94% in China [9,10,11].

There are limited studies on vaccine hesitancy among older adults in the community. Current studies on this subject show that vaccine hesitancy is a universal problem. Fear of the side effects of the vaccine, doubts about its safety, and doubts about its necessity and efficacy are generally stated as the most prevalent factors in vaccine hesitancy [12].

Vaccine indecision is a growing problem in Turkey despite the high coverage of childhood vaccines. In the studies carried out, in addition to the universal reasons for the hesitancy against the vaccine in Turkey, there are also religious beliefs which lead to distrust in the ingredients [13,14]. Therefore, it is necessary to investigate the reasons behind the hesitation to receive the COVID-19 vaccine and its effective immunogenicity, which can help us to understand the determinants of vaccine uptake.

Considering the above reasons, this study was carried out to determine the factors that influence individuals between the ages of 65 and 75 to not receive the third booster dose, to advise those who are hesitant, and to learn their thoughts about taking the third dose.

## 2. Material and Methods

### 2.1. Type of Research

This study is a cross-sectional study using a telephone-based survey design.

### 2.2. Study Population

This study was conducted between April and May 2022 among 2383 older adults aged between 65–75 who had never received the COVID-19 booster dose, according to the records of the District Health Directorate in the Sultanbeyli district of Istanbul. The total population of the Sultanbeyli district is 349,485 and the population between the ages of 65–75 consists of 9060 people, comprising 2.5%. Another feature of the Sultanbeyli district is that it is the lowest district of Istanbul in the socio-economic development index [15]. The sample size was not calculated in the study, as we aimed to reach all older people between the ages of 65–75 who did not receive the COVID-19 booster dose. The reason we worked with older adults is that they are both at risk for COVID-19 and open to guidance by their caregivers. There is much misinformation among older adults, especially those with chronic diseases, about not being vaccinated which states that they will experience side effects. These people are also a population open to manipulation, and they can be easily influenced. It is important that healthcare personnel can access the right sources and information and that they can be effective in their decisions. We found that 103 of the participants could not be reached because they did not have a registered phone number in the system or because the wrong number was registered, and 798 of them could not be reached because they did not answer the phone. In addition, 196 older adults with foreign nationalities were not included in the study because they could not be contacted due to the language barrier, and 211 older adults refused to participate in the study. The study was completed with 1075 older adults.

### 2.3. Measuring Tools

A questionnaire prepared by researchers, consisting of three parts, was given to older adults via telephone. Before the application of the questionnaire, training was given to the people who would make the phone calls to older adults. In the first part of the questionnaire, socio-demographic questions such as age, education level, employment status, income level, smoking status, and any chronic diseases were included. The income level of individuals was determined based on 4682TL, which is the hunger limit of a single person at the time of data collection. Those whose income was above 4682TL were considered above the hunger limit. In the second part, questions about having a COVID-19 infection, history of hospitalization, and the risk of contracting COVID-19 were asked. In the last section, it was asked which COVID-19 vaccine they had received before, whether there were other vaccines that the older adults should receive in the event of any side effects of the COVID-19 vaccines, what side effects they had, and their reasons for not receiving a booster dose. At the end of the survey, the participants were briefed about the importance of COVID-19 vaccination and booster doses, and they were asked whether they would be vaccinated afterward. According to this, the answers, given as “I definitely will”, “I am unsure”, or “I definitely will not” were evaluated.

### 2.4. Statistical Analysis

Descriptive data are presented as means, standard deviation values, and frequency tables. For statistical analysis of the data, the Chi-square test was used to compare variables. The conformity of the variables to the normal distribution was examined using visual (histogram) and analytical methods (Kolmogorov–Smirnov/Shapiro–Wilk). Logistic regression analysis was performed to control the confounders of the variables found to be significant in univariate analyses. In this study, *p* < 0.05 was considered statistically significant.

### 2.5. Ethical Considerations

Before the study, ethics committee approval and research permits were obtained from the Medipol University Ethics Committee, with a protocol number of 339, dated 13 April 2022. The subjects in the research were asked to participate in the study after being informed about the research and its permits. Our study was conducted according to the Declaration of Helsinki, and written informed consent was obtained from all participants.

## 3. Results

This research was completed with 1075 participants, reaching 45% of people aged 65–75 who did not receive the third dose of the COVID-19 vaccine in the region.

The most frequently reported reasons for not receiving a booster dose by the study participants are shown in detail in Table 1. It was stated by some respondents that the third dose was ineffective or unnecessary. As for other reasons, it was stated that they went to get vaccinated, but they were turned away because the required number of people for vaccination was not reached. Their thoughts on getting vaccinated in the upcoming period were compared according to these reasons.

In this study, 64.2% of the participants were female and 35.8% were male, and the mean age was 69.33 ± 2.88. Considering the distribution of the participants according to their educational status, 52.0% were uneducated.

When the relationship between people’s attitudes towards getting vaccinated and their educational status was compared, it was found that educated people were significantly more likely to be vaccinated (*p* < 0.001). When evaluated according to per capita income, the monthly income level of 74.2% of the participants was below 4682 TL, which is the hunger limit. Although 69.6% of the participants had a chronic disease, the most reported chronic diseases were hypertension, diabetes, hyperlipidemia, and cardiological and respiratory diseases.

When the participants were asked how they evaluated their health, 53.3% rated it as good. At the end of the questionnaire, a speech was given to the participants about the importance of the booster dose and the need for it, and as a result, they were asked about their thoughts on receiving a booster dose in the upcoming period. Regarding this, 33.4% of them stated that they definitely would and 66.6% of them stated that they were unsure or they would not. The relationships between the sociodemographic characteristics of the participants and their intention to receive a booster dose are shown in detail in Table 2.

When asked about influenza and pneumococcal vaccines, which are the vaccines that the participants should have due to their age, 89% stated that they did not have a flu vaccine and 94.5% stated that they did not have a pneumococcal vaccine. When asked whether they encountered any side effects from the first two doses of the COVID-19 vaccine, 29.5% stated that they experienced side effects, and the most frequently reported side effects were arm pain, weakness, and fever. The rate at which they reported side effects to the relevant institutions was 8.1% (Table 3).

In our logistic regression analysis, the opinion about getting vaccinated, which is the dependent variable, was divided into two groups: those who said they would, and those who were unsure or who said they would not. Logistic regression analysis was also performed with the variables found to be significant in univariate analyses. The variables included in the logistic regression analysis model were previous vaccination with the influenza vaccine; which COVID-19 vaccine they received previously; educational status; side effects; and neglect, redundancy, ineffectiveness, and lack of time as reasons for hesitancy. Those who stated that they had received an influenza vaccine before were significantly 1.9 times (95% CI 1.22–2.99) more likely to be vaccinated with a booster dose. It was determined that those who stated that they were not vaccinated due to lack of time and forgetting/neglect tended to get vaccinated at a rate 1.4 and 5.6 times higher, respectively. According to their educational status, those who were uneducated reported that they would get vaccinated 0.5 times (95% CI 0.42–0.76) less often than those who were educated. (Table 4).

## 4. Discussion

Vaccination hesitancy is an important variable in the entire population, but it is more important in populations with risk groups. In older adults, who constitute a risk group, this situation requires extra attention and awareness, as it can have an impact on them. This study has shed light on this awareness by working with older adults, who are a risk group, and with those who had not been vaccinated with the COVID-19 booster dose.

In this study, no significant relationships were found between the sociodemographic characteristics of older adults, such as gender, income status, marital status, and presence of a chronic disease, and booster dose intake. The income levels of the participants did not affect their access to the vaccine, which was the expected result. On the other hand, those with chronic diseases are more susceptible, and they may have received more vaccines because of their fear of the effects of COVID-19. Conversely, the rate of rejection may have increased due to fear of the side effects of the vaccine. In this study, this distinction was not fully seen, and no difference was found. In future qualitative studies, the effect of having a chronic disease on vaccination can be investigated in more detail. A study in Algeria found that older adults and those with chronic illnesses were more likely to accept a booster dose than younger people [16].

During the study period, all of the participants were vaccinated with at least one dose of the COVID-19 vaccine, and none of them were vaccinated with a booster dose. Since the Sultanbeyli region is the lowest-ranked socioeconomic region in Istanbul, the education levels of the participants were not high. Education levels were categorized according to whether they were educated or not, and this factor was found to be significantly related to the idea of receiving a booster dose. Similarly to this study, in a study conducted in China, a higher education level was found to be positively associated with booster dose uptake of the COVID-19 vaccine [17]. In another study conducted with factory workers in China, it was found that those with higher education levels were more likely to receive a booster dose of the COVID-19 vaccine [18]. In this study, pneumococcal and flu vaccine history were investigated as variables that may have affected willingness to receive the COVID-19 vaccine booster dose, and it was found that those who had received the flu vaccine indicated that they were significantly more certain about the booster dose. Previous studies have found that pneumococcal vaccination history was associated with higher uptake of the primary COVID-19 vaccine series among older adults in Hong Kong, Italy, Canada, and Saudi Arabia [17,19,20,21].

These findings may be explained by the stronger motivation of older adults with pneumococcal and flu vaccination experience to get their vaccinations to prevent infectious diseases. In this study, people were asked about the vaccine types they were given in the first two doses, and it was found that those who had received the Pfizer-BioNTech vaccine refused significantly more booster doses. Because the mRNA technology used to create the Pfizer-BioNTech vaccine is new, its consequences are more difficult to predict and more difficult to understand and accept, especially in the older population [22]. Additionally, in this study, those who received the Pfizer-BioNTech vaccine instead of the Sinovac vaccine reported more side effects, such as fever and arm pain. Additionally, in this study, it was determined that those who experienced side effects from their previous vaccines stated that they had significantly less intention to receive a booster dose.

We must recognize a practical problem, namely that the longer it takes to achieve herd immunity, the longer it will take for normal economic development to continue, and the more people are likely to face more complex situations [11,23,24].

In addition, vaccination is a natural choice for establishing herd immunity, especially in high-risk groups such as older adults, to provide longer-lasting immunity and greater protection against variants of COVID-19. The benefits of vaccinating older adults for their condition far outweigh the risks. Therefore, reasons for hesitation regarding vaccination among older adults should realistically be reduced, and we should seek to resolve them. It is very important to simplify the vaccination process, to know the reasons behind hesitation, and to develop solutions. In this study, in addition the hesitations of the elderly about getting vaccinated, their problems with reaching vaccination were also revealed. Our findings will assist in evaluating the attitudes of older adults towards booster doses and investigating the associated factors influencing vaccination behaviors; this may provide theoretical and practical implications for subsequent immunization strategies in older populations.

## 5. Strengths and Limitations

This study provides important information about the factors associated with COVID-19 vaccine hesitancy, as well as vaccine hesitancy that may affect vaccine uptake among the elderly population. A notable strength of the study is that the sample is nationally representative.

In addition, the study has an important role in the fact that it was conducted on elderly individuals who had only received two vaccines and were hesitant to take the booster dose. Another strength of this study is that although it is cross-sectional, it also has a quasi-experimental feature, since it also evaluates the vaccination decisions according to the information about vaccination that people provided when they were called. This study is also limited by other factors. First, it was conducted in a single center and included only older adults who had not received a booster dose of COVID-19. Additionally, we could not reach all of them due to the reasons that we mentioned in the methods section. Next, data were collected only based on the participants’ declarations, and it could not be confirmed whether those who stated that they would be vaccinated were vaccinated.

## 6. Conclusions

In this cross-sectional study, 33.7% of older adults said they would receive a booster dose of the COVID-19 vaccine, while 40% were undecided. Acceptance behaviors were closely related to education, experiencing side effects, previous vaccination status, and attitudes toward other vaccines. Concerns about contraindications, vaccine safety, and limited mobility were the main reasons for vaccine hesitancy.

In the current situation, it is also very important to state the fact that valid immunity among the older population is not only necessary to prevent infection, severe illness, and death caused by emerging variants, but also to reduce the enormous disease and economic burden caused by the long-term sequelae of COVID-19. In our study, the level of education was an effective factor in determining whether people would receive a booster dose. Health communication messages promoting COVID-19 booster doses among older adults should be simple and easy to understand for people with low literacy levels. In addition, one of the most effective reasons for not receiving a booster dose is that people experienced side effects from previous vaccination. This study showed that it is possible to change people’s minds after they are told about the importance of getting vaccinated, and they are mostly not vaccinated due to a lack of information. We believe that it is important to vaccinate older patients; further, since immunity conferred by vaccination may decline over time, mortality rates decrease with the administration of additional doses.

## Figures and Tables

**Table 1 vaccines-11-00542-t001:** Distribution of the reasons why the participants did not receive the booster dose of COVID-19.

	Intention to Receive Booster Vaccination		*p*-Value
	Definitely Yes n (%)	Unsure or Definitely No n (%)	
Side effect fear	33 (3.8)	81 (9.3)	**<0.001**
Redundancy	68 (7.8)	154 (17.7)	**<0.001**
Have had COVID-19	43 (4.9)	74 (8.5)	0.385
Ineffective	93 (10.7)	208 (23.9)	**<0.001**
Forget/neglect	9 (1.0)	66 (7.6)	**<0.001**
Waiting for Turkovac	9 (1.0)	15 (1.8)	0.122
Lack of time	12 (1.4)	21 (2.4)	**<0.001**
Didn’t know it had to be	13 (1.5)	21 (2.4)	0.092
Have a chronic disease	11 (1.3)	27 (3.1)	0.587
Have a disability	25 (2.9)	60 (6.9)	0.941
Other reasons	15 (1.7)	26 (2.9)	0.765

Bold values are significant, with *p* < 0.05. CI− and CI+ are the lower and upper bonds of the 95% confidence interval.

**Table 2 vaccines-11-00542-t002:** Characteristics of participants by intention to receive the booster vaccination.

Characteristics		Intention to Receive Booster Vaccination		*p*-Value *	
		Definitely Yes n (%)	Unsure or Definitely No n (%)		
Sex	Woman	226 (32.8)	464 (67.2)	0.392	
	Man	136 (35.3)	249 (64.7)		
Education status	Uneducated	160 (28.6)	399 (71.4)	**<0.001**	
	Educated	202 (39.2)	314 (60.8)		
Income status	Lower than 4682 TL	284 (35.7)	513 (64.3)	0.064	
	Higher than 4682 TL	50 (27.1)	134 (72.8)		
	Don’t know	28 (29.8)	66 (70.2)		
Marital status	Married	246 (34.7)	463 (64.3)	0.360	
	Divorced or widowed	105 (31.0)	234 (69.0)		
	Single	11 (40.7)	16 (59.3)		
Employment	Yes	17 (38.7)	27 (61.3)	0.477	
	No	345 (33.5)	686 (66.5)		
Chronic disease	Yes	263 (35.1)	485 (64.9)	0.119	
	No	99 (30.4)	228 (69.6)		
Self-health assessment	Very bad	4 (30.8)	9 (69.2)	0.407	
	Bad	37 (43.1)	49 (56.9)		
	Middle	111 (32.4)	233 (67.6)		
	Good	192 (34.0)	381 (66.0)		
	Very good	18 (30.5)	41 (69.5)		
Smoking	Yes	37 (38.3)	60 (61.7)	0.329	
	No	325 (33.3)	653 (66.7)		
Age			Mean ± sd		
			69.33 ± 2.88		

Bold values are significant, with *p* < 0.05. CI− and CI+ are the lower and upper bonds of the 95% confidence interval. * Chi-square test.

**Table 3 vaccines-11-00542-t003:** Characteristics of participants related to other vaccines by intention to receive the booster vaccination.

		Intention to Receive Booster Vaccination		*p*-Value *
		Definitely Yes n (%)	Unsure or Definitely No n (%)	
Ever had flu vaccine	Yes	51 (43.0)	68 (57.0)	**0.025**
	No	311 (32.5)	645 (67.5)	
Ever had pneumonia vaccine	Yes	26 (44.1)	33 (55.9)	0.082
	No	336 (33.2)	680 (66.8)	
Which COVID-19 vaccine was received before	Pfizer-BioNTech	131 (29.1)	319 (70.9)	**0.018**
	Sinovac	191 (38.5)	304 (61.5)	
	Don’t know	34 (30.6)	77 (69.4)	
	Pfizer-BioNTech and Sinovac	6 (31.5)	13 (68.5)	
Any side effects due to the COVID-19 vaccine	Yes	98 (31.0)	219 (69.0)	0.216
	No	264 (34.8)	494 (65.2)	
Reported any side effects (n = 317)	Yes	5 (19.2)	21 (80.8)	0.684
	No	83 (28.5)	208 (71.5)	

Bold values are significant, with *p* < 0.05. CI− and CI+ are the lower and upper bonds of the 95% confidence interval. * Chi-square test.

**Table 4 vaccines-11-00542-t004:** Comparison of variables with logistic regression analysis.

	*p*-Value	OR	95% C.I. for EXP(B)	
			Lower	Upper
Ever had influenza vaccine (ref: no)	** 0.004 **	1.917	1.226	2.998
Which COVID-19 vaccine was received before (ref: don’t know)	<0.001			
Which COVID-19 vaccine was received before (Biontech)	0.551	0.858	0.518	1.420
Which COVID-19 vaccine was received before (Sinovac)	** 0.016 **	1.825	1.119	2.975
Which COVID-19 vaccine was received before (Biontech and Sinovac)	0.854	1.117	0.344	3.630
Any side effects (ref: no)	** <0.001 **	0.119	0.064	0.220
Lack of time (ref: no)	** 0.044 **	1.414	1.010	1.982
Redundancy (ref: no)	** <0.001 **	0.235	0.164	0.337
Ineffectiveness (ref: no)	** 0.004 **	0.051	0.007	0.380
Forget/neglect (ref: no)	** <0.001 **	5.623	2.582	12.246
Education (ref: literate)	** <0.001 **	0.571	0.428	0.762
Constant	**0.007**	0.167		

Bold values are significant, with *p* < 0.05. CI− and CI+ are the lower and upper bonds of the 95% confidence interval.

## Data Availability

The corresponding author had full access to all the data in the study.

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
