# Peer review of "COVID-19 Vaccine Booster Dose Acceptance among Older Adults"

_vaccines, 2023, doi:10.3390/vaccines11030542_

Round 1

Reviewer 1 Report

General comments

Thank you for the opportunity to review this manuscript describing a cross- sectional study examining the factors driving booster vaccination hesitancy among those over 65 residing within a particular district of Turkey.

The objective of the study is clearly stated and pertinent. The study seems appropriately designed to achieve this objective.

Major comments

Results

1.       Suggest including univariate odds ratios of “definitely yes” versus “unsure or definitely no” as well as P values in Tables 1-4. It makes sense to define “vaccine hesitancy” as those who are both “unsure or definitely no” because this is the group that from a practical, public health point of view, might benefit from a more targeted and informed booster promotion campaign.     Confidence intervals would also be helpful. This simple analysis will help give the reader a better appreciation of the magnitude of each particular effect on vaccine hesitancy (p values give no information about the effect size). Raw numbers may be included in tables or could potentially be included in supplemental material (if this is an option provided by journal). Wherever possible throughout analysis suggest consistently using “definitely yes” as reference (ie the “good outcome” of interest). This will mean OR significantly associated with booster hesitancy will have values less than one (and an upper CI < 1). By avoiding double negative expressions I suspect this approach will make interpretation conceptually simpler for the reader (eg expressions such as “less likely to not seek vaccination” could then be rephrased as “less likely to seek vaccination”). 

2.       Consider repeating logistic regression including those factors statistically significant on univariate analysis using approach above. This approach makes sense because it means the same comparator groups are used in both the univariate and multivariate analyses.

Abstract

1.       In results section include OR and confidence intervals for each significant factor (on multivariate analysis) to give information about effect size (once again with “definitely yes” as the good outcome of interest, such that factors associated with hesitancy will have OR less than one).

2.       Conclusion must relate to study findings and not simply restate a priori assumptions – eg vaccination is important. Conclusions should state in very broad terms how the study findings should inform future campaigns to decrease vaccine hesitancy in the over 65 population.

Methods

In many parts of the world misinformation and disinformation posted on the internet has played some role in vaccine hesitancy. Is there any information on internet access for the population studied and misinformation about vaccines presented in an understandable format to the population studied? Would be worth commenting generally on this background even if not specifically asked about in the survey. This information may also be mentioned in introduction and discussion.

Minor comments

1.       Introduction –appropriately explains rationale for study. Very minor suggestions - suggest using expression “vaccine hesitancy” in line 63. Line 60 suggest using “rate of uptake” rather than rate of receiving”.

2.       Material and Methods – Line 72; I don’t think this is a classic “quasi experimental study” so suggest removing this. Quasi experimental studies generally refer to “before-after interventional studies”. Suggest simply describing as a cross sectional study using a telephone- based survey design. This gives the reader a quick overview /synopsis of the study design.  Lines 96-99 suggest rewriting: “The income level of individuals was determined based on 4682TL, which is the hunger limit of a single person at the time of data collection. Those whose income is above 4682TL are considered above the hunger limit.”

3.       Results – Line 171 suggest rewriting “…to side effects were X times as likely to seek vaccination.” Where X = univariate OR calculated as suggested above.  Better to include OR as indicators of effect size where available and helpful to express consistently such that OR<1 associated with vaccine hesitancy. Lines 181-182 suggest rewriting as “Those who did not have an influenza before were X as likely to seek vaccination.” Where X = univariate OR calculated as suggested above. Similarly change lines 183-184, expressing OR to reflect likelihood of seeking vaccination.

4.       Discussion – Line 201 – change “intake” to “uptake”. Line 193 change “against” for “with”.

Author Response

Response to Reviewer 1 Comments

Point 1: Results

  1. Suggest including univariate odds ratios of “definitely yes” versus “unsure or definitely no” as well as P values in Tables 1-4. It makes sense to define “vaccine hesitancy” as those who are both “unsure or definitely no” because this is the group that from a practical, public health point of view, might benefit from a more targeted and informed booster promotion campaign.     Confidence intervals would also be helpful. This simple analysis will help give the reader a better appreciation of the magnitude of each particular effect on vaccine hesitancy (p values give no information about the effect size). Raw numbers may be included in tables or could potentially be included in supplemental material (if this is an option provided by journal). Wherever possible throughout analysis suggest consistently using “definitely yes” as reference (ie the “good outcome” of interest). This will mean OR significantly associated with booster hesitancy will have values less than one (and an upper CI < 1). By avoiding double negative expressions I suspect this approach will make interpretation conceptually simpler for the reader (eg expressions such as “less likely to not seek vaccination” could then be rephrased as “less likely to seek vaccination”).  

  1. Consider repeating logistic regression including those factors statistically significant on univariate analysis using approach above. This approach makes sense because it means the same comparator groups are used in both the univariate and multivariate analyses. 

Response 1: Thank you very much for your valuable comment. Tables 1-4 have been revised. Table 2 has been removed and the others have been adjusted for definitely yes” versus “unsure or definitely no” .You can see the corrections in the main manuscript that we uploaded.

Point 2: Abstract

  1. In results section include OR and confidence intervals for each significant factor (on multivariate analysis) to give information about effect size (once again with “definitely yes” as the good outcome of interest, such that factors associated with hesitancy will have OR less than one). 
  2. Conclusion must relate to study findings and not simply restate a priori assumptions – eg vaccination is important. Conclusions should state in very broad terms how the study findings should inform future campaigns to decrease vaccine hesitancy in the over 65 population. 

Response 2: Thank you for your suggestion. As you suggest, we revised and adjusted Abstract, we add some important information about our results to the conclusion part. We wrote OR and CI for each significant factor. We hope it seems better now.

Point 3: Methods

In many parts of the world misinformation and disinformation posted on the internet has played some role in vaccine hesitancy. Is there any information on internet access for the population studied and misinformation about vaccines presented in an understandable format to the population studied? Would be worth commenting generally on this background even if not specifically asked about in the survey. This information may also be mentioned in introduction and discussion. 

 Response 3: Thank you for your suggestion. We added some extra information that shows why we worked on this population and if there is any misinformation. You can see the corrections in the main manuscript.

Minor comments

  1. Introduction–appropriately explains rationale for study. Very minor suggestions - suggest using expression “vaccine hesitancy” in line 63. Line 60 suggest using “rate of uptake” rather than rate of receiving”.
  2. Material and Methods– Line 72; I don’t think this is a classic “quasi experimental study” so suggest removing this. Quasi experimental studies generally refer to “before-after interventional studies”. Suggest simply describing as a cross sectional study using a telephone- based survey design. This gives the reader a quick overview /synopsis of the study design.  Lines 96-99 suggest rewriting: “The income level of individuals was determined based on 4682TL, which is the hunger limit of a single person at the time of data collection. Those whose income is above 4682TL are considered above the hunger limit.” 
  3. Results – Line 171 suggest rewriting “…to side effects were X times as likely to seek vaccination.” Where X = univariate OR calculated as suggested above.  Better to include OR as indicators of effect size where available and helpful to express consistently such that OR<1 associated with vaccine hesitancy. Lines 181-182 suggest rewriting as “Those who did not have an influenza before were X as likely to seek vaccination.” Where X = univariate OR calculated as suggested above. Similarly change lines 183-184, expressing OR to reflect likelihood of seeking vaccination. 
  4. Discussion – Line 201 – change “intake” to “uptake”. Line 193 change “against” for “with”. 

Response 5: Thank you for your minor comments. We made all the revisions that you suggested us. You can see the corrections in the main manuscript.

Reviewer 2 Report

The factors responsible for hesitancy to receive COVID-19 vaccine boosters against emerging variants is an important topic and worthy of publication. However, this presentation needs marked improvement, especially in light of the results, which showed that the apparent educational status and fear of side-effects were the major factors responsible for the hesitancy.

The term illiteracy should be replaced by educational status, and the cutoff of educational status be defined.

The questionnaire itself should be displayed.

Figure 1 is acceptable, but not necessary; the results could be simply stated in the text.

Table 1 should list the reasons for hesitancy (as listed in Table 4) according to the characteristics listed in current Table 1. Not sure why bowel cancer screening is an important characteristic, and should be eliminated.

Table 2 can be eliminate, as the result is predictable and can be stated in the text.

A new Table that lists the only significant factors regarding the intention to receive a booster vaccine could replace the other existing Tables.

The discussion should include what factors among the messages given to the study participants were important to the participants willingness to receive the booster vaccines.

Author Response

The term illiteracy should be replaced by educational status, and the cutoff of educational status be defined.

The questionnaire itself should be displayed.

Figure 1 is acceptable, but not necessary; the results could be simply stated in the text.

Table 1 should list the reasons for hesitancy (as listed in Table 4) according to the characteristics listed in current Table 1. Not sure why bowel cancer screening is an important characteristic, and should be eliminated.

Table 2 can be eliminate, as the result is predictable and can be stated in the text.

A new Table that lists the only significant factors regarding the intention to receive a booster vaccine could replace the other existing Tables.

The discussion should include what factors among the messages given to the study participants were important to the participants willingness to receive the booster vaccines.

Response: Thank you very much for your comments and suggestions. We removed Figure 1, table 2 and bowel cancer characteristic as you suggested. We categorized educational status as educated and uneducated. Because there were lots of individuals that never went school. Questionnaire will be displayed by corresponding author when asked. You can see all the corrections in the main manuscript. We hope it looks better now.

Round 2

Reviewer 2 Report

The authors have responded to most of the criticisms and suggestions. However, the paper could be improved by making Table 4 Table 1, then proceeding to indicate the factors underlying decisions regarding the intention to receive booster vaccines.

Author Response

Point 1: However, the paper could be improved by making Table 4 Table 1, then proceeding to indicate the factors underlying decisions regarding the intention to receive booster vaccines.

Response 1: Thank you very much for your contributions. First of all, we made some additions to increase our word limit to over 4000. Afterwards, we made the arrangements for Tables 1 and 4 in line with the minor revision you provided. You can see the changes made in the main manuscript.
